# Heartbeat Classification and Arrhythmia Detection Using a Multi-Model Deep-Learning Technique

**DOI:** 10.3390/s22155606

**Published:** 2022-07-27

**Authors:** Saad Irfan, Nadeem Anjum, Turke Althobaiti, Abdullah Alhumaidi Alotaibi, Abdul Basit Siddiqui, Naeem Ramzan

**Affiliations:** 1Department of Computer Science, Capital University of Science and Technology, Islamabad 44000, Pakistan; mcs191026@cust.pk (S.I.); abasit.siddiqui@cust.edu.pk (A.B.S.); 2Faculty of Science, Northern Border University, Arar 1321, Saudi Arabia; turke.althobaiti@nbu.edu.sa; 3Department of Science and Technology, College of Ranyah, Taif University, Taif 11099, Saudi Arabia; a.alhumaidi@tu.edu.sa; 4School of Computing, Engineering and Physical Sciences, University of the West of Scotland, Paisley PA1 2BE, UK; naeem.ramzan@uws.ac.uk

**Keywords:** feature extraction, cardiac arrhythmia, ECG classification, hybrid models, deep learning

## Abstract

Cardiac arrhythmias pose a significant danger to human life; therefore, it is of utmost importance to be able to efficiently diagnose these arrhythmias promptly. There exist many techniques for the detection of arrhythmias; however, the most widely adopted method is the use of an Electrocardiogram (ECG). The manual analysis of ECGs by medical experts is often inefficient. Therefore, the detection and recognition of ECG characteristics via machine-learning techniques have become prevalent. There are two major drawbacks of existing machine-learning approaches: (a) they require extensive training time; and (b) they require manual feature selection. To address these issues, this paper presents a novel deep-learning framework that integrates various networks by stacking similar layers in each network to produce a single robust model. The proposed framework has been tested on two publicly available datasets for the recognition of five micro-classes of arrhythmias. The overall classification sensitivity, specificity, positive predictive value, and accuracy of the proposed approach are 98.37%, 99.59%, 98.41%, and 99.35%, respectively. The results are compared with state-of-the-art approaches. The proposed approach outperformed the existing approaches in terms of sensitivity, specificity, positive predictive value, accuracy and computational cost.

## 1. Introduction

Cardiac arrhythmia is categorized as the irregular beating of the heart [1]. This irregularity may either be a slow or fast heartbeat. A heart rate of over 100 beats per minute (bpm) is categorized as tachycardia, while the instance of a pulse lower than 60 bpm is alluded to as bradycardia. Global statistics reveal that a significant population suffers from heart diseases which manifest in the form of heart attacks, strokes, etc.; furthermore, these afflictions are one of the significant reasons for death all over the planet. Moreover, treatment for heart diseases is too costly, and only a limited number of patients have the luxury of affording it [2].

Electrocardiograms (ECGs) are designed to analyze arrhythmias. ECG is used to monitor the functioning of the heart by capturing electrical activity [3]. ECG is based on a wave-like feature that mainly includes the P, QRS, and T waves. Furthermore, it accumulates 12 lead signals that are generated from the cords attached to the patient’s body. These signals are divided into six limb-based electrodes configuration i.e., aVR, aVL, aVF, I, II, III, and six chest-based electrodes configuration i.e., V1, V2, V3, V4, V5, V6. Each electrode measures a stream of electrical signals generated by the heart from a different angle covering both the horizontal and vertical planes [4].

In the interim, progress, as far as accessible computational devices and algorithms are concerned, has uncovered their use in automated detection techniques. As a result, diagnosis of cardiovascular anomalies is on the ascent. As of late, attention to ECG beat and rhythm characterization has also been on the rise [5]. ECG arrangement can be characterized into segments that emphasize on tracking down viable feature-extraction strategies, further improving the classification results, and use of machine-learning techniques (ML) to improve the accuracy of these strategies such as Decision Trees (DT) [6,7,8], K-Nearest Neighbor (KNN) [9,10,11], Linear Support Vector Machines (SVM) [12,13,14,15] and Random Forest (RF) [16,17], etc.

The volume of data related to cardiac arrhythmias has expanded to an exceptional level, in recent years, which had limited the improvements in feature-extraction results. To that end, deep learning (DL) has managed to achieve vital outcomes in the domain of arrhythmia detection. The key characteristic of deep neural networks involves the automated process of feature detection and extraction in providing concise and accurate results, which thusly delivered an allure in the space of heartbeat classification [18]. A plethora of techniques and methods have been incorporated with state-of-the-art deep-learning algorithms to fully use the potential of automated feature recognition and extraction. Such techniques are not only based on uni-model frameworks but multi-model and hybrid frameworks as well. The hierarchical layered structure in the deep neural network integrates multi-level features and their transformation. This structure further helps in the refinement of features [19].

To accomplish the aforementioned tasks, neural networks such as Recurrent Neural Networks (RNN) [20,21,22], Long Short-Term Memory (LSTM) [23,24], Convolutional Neural Networks (CNN) [25,26,27], as well as hybrid models [28,29,30,31] etc., are being integrated to overcome the hindrances of conventional machine-learning strategies that were subject to manual and inaccurate selection of features that may incite inconvenient impacts for the current applications. The drawbacks of the hybrid approaches accumulate the increasing cost and lack of quality datasets which, however, can be considered negligible in some viable cases because the precise classification of heartbeats along with the accurate detection of arrhythmia requires a substantial amount of data to work with [32].

To address these issues at hand, this work proposes an ensemble of deep neural networks that involves the designing and merging of two neural networks followed by the training of the merged model in a simultaneous flow. The novel aspects highlight the implementation of a multi-model framework that incorporates the ability to merge multiple ML/DL models and produce a robust output. The proposed framework has shown superior results in heartbeat detection and classification compared to state-of-the-art works.

The further sections have been segmented as follows: Section 2 gives an itemized survey of the literature; Section 3 portrays the proposed methodology; Section 4 expresses the trial examination and depicts the correlation of results with the modern approaches; Section 5 concludes the following research work, and Appendix A incorporates the GitHub repository link for the source code of the proposed approach.

## 2. Related Works

The literature review has been partitioned into two subsections, (i) conventional machine-learning approaches, and (ii) deep-learning/hybrid approaches.

### 2.1. Machine-Learning Approaches

AI assumes a crucial part in medical prediction and grouping in the clinical area. It gives colossal assistance to the doctors to deal with an enormous measure of captured clinical information. These strategies can assist with the early and better finding of illnesses that can save the medicinal expenses and costly clinical trials [33]. Gupta et al. [34] demonstrated the implementation of multiple machine-learning algorithms which included naïve Bayes, random forest, SVM, etc. An implementation of the learner module using a linear SVM and Random Forest was also proposed. The model was tested on the publicly available MIT-BIH datasets and returned a classification accuracy of 77.4%. The proposed model not only showed slightly better classification results but a decreased training time as well.

Luz et al. [35] implemented six feature-extraction models for the comparative analysis of different performance metrics. An optimal path forest classifier (OPF) was designed, which, in terms of accuracy, was not superior to a fine-tuned SVM; however, the overall model training time was greatly reduced. Sarfraz et al. [36] used the Independent Component Correlation (ICC) algorithm along with the generic functions of ECG for pattern detection and recognition. The features extracted by the ICCA were divided into training and testing tests. The accompanying methodology showed great outcomes regarding accuracy and precision; however, due to the manual feature detection and extraction, it proved costly and difficult to put into practical use.

Batra and Jawa [37] combined gradient boosting with SVM for the efficient detection of arrhythmia from ECGs. The proposed approach was benchmarked with other machine-learning algorithms such as random forest, gradient boosting, decision trees, etc. Before the final training of models, the raw data had undergone extensive processing and feature selection processes. The model achieved an overall recognition rate of 84.82%. Namrata and Pradeep [38] came up with a novel feature selection technique that worked on the principle of best-first selection. The three-filter feature selection (TFFS) approach filtered out the subset of optimal features from the publicly available MIT-BIH dataset. The final input was fed to three classifiers, namely JRip, SVM, and random forest. The comparative results depicted the superior performance of random forest with a classification accuracy of 85.58%.

Miquel et al. [39] proposed an Echo State Network (ESN) classifier for the classification of heartbeats. The proposed model was capable of producing accurate results while needing only a single ECG lead. A combination of multiple ensembles was also demonstrated which exploited the parallelism for additional training speed. The proposed approach was tested on two publicly available datasets and demonstrated the highest accuracy of 98.6% on lead II. An extensive amount of preprocessing and data cleaning was involved in the overall methodology which ultimately increased the cost of the proposed approach.

Therefore, this section concludes that even though machine learning has paved the path for tremendous progress in the domain of medicine and health automation systems, the traditional machine-learning algorithms still suffer from the problems of manual feature recognition, the curse of dimensionality, and overfitting, etc. To that end, state-of-the-art trends have shifted towards the automation of feature recognition by using deep neural networks.

### 2.2. Deep-Learning/Hybrid Approaches

Over the last few years, deep learning has accomplished astounding outcomes in the space of Artificial Intelligence (AI). One such aspect is the application of deep-learning methods in healthcare. Providing medical centers with automated health-maintenance technologies has revolutionized the field of health care by minimizing the cost while increasing the efficiency of the services. A substantial amount of research has been done in the domain of cardiac healthcare.

Gao et al. [40] proposed an effective LSTM-based approach to identify eight heartbeats. Several condensed LSTM layers were stacked for the accurate detection of arrhythmias; however, the proposed approach required a colossal pool of labeled data. Furthermore, due to the deeply layered architecture, the model required a long time to train and produce noteworthy results. Amrita and Kyung [41] classified the time-series sequence of ECG data relating to the normal and abnormal beating of the heart. An exclusive LSTM layer-based classifier was designed with an emphasis on design simplification. The classifier successfully managed to recognize five arrhythmias with an accuracy of 95%. The paper has not, however, discussed nor taken into consideration the overall training time of the classifier which assumes a significant part in limiting the general expense of the approach.

Hiriyannaiah et al. [42] provided a detailed analysis of multiple deep LSTM models to capture temporal dependencies from ECG signals. The performance of four stacked LSTM (3 LSTM, 1 BiLSTM) models was inter-compared. The benchmark statistics based on publicly available datasets revealed that the bidirectional LSTM-based model achieved the highest accuracy of 95% compared to all-LSTM stacked models. The training time per epoch, however, was greatly increased on the implementation of bidirectional LSTM which resulted in an increased computational cost of the approach.

Parvaneh et al. [43] reviewed the traditional deep-learning algorithms which included CNN, RNN, Auto-Encoders, and Deep Belief Networks. The paper emphasized the advancements and benefits of deep neural networks over shallow machine-learning techniques. A statistical review regarding the most widely used neural network for heartbeat and arrhythmia classification was also carried out. The statistics revealed that the convolutional neural networks (CNN) had been integrated by most research works to date. The paper additionally underscored the qualities of CNN that made it feasible for being the most ordinarily used in sequence to label classification problems. Acharya et al. [44] classified five different heartbeats with a deeply layered convolutional neural network. To overcome the class imbalance problem in the original data, synthetic data were generated based on a few resampling techniques. The experimental analysis revealed the model’s ability to achieve a classification accuracy of 94% with noise removal and 93.5% without any preprocessing.

Another approach [45] proposed a deep neural network (DNN)-based framework to ameliorate the difficulties faced while detecting arrhythmias. The approach featured a learning stage in conjunction with a robust feature-extraction protocol. The combined subset of the most optimal features was aggregated with the help of a genetic algorithm. The framework was designed as an analytic module for the detection of anomalies in certain medical conditions. The methodology accomplished a 94.00% accuracy in the classification of five arrhythmia super-classes. Chen et al. [46] classified nine arrhythmia micro-classes using an ECG dataset with 12-lead signals. The approach accumulated a 1D deep convolutional neural network that also secured the top spot in the China Physiological Signal Challenge (CPSC). With relevance to the clinical observations and the data available, the proposed model reported an average recognition rate of 97%. The benchmark against the referenced approaches was carried out based on the F1 score which was calculated to be 0.84.

In more recent work, Wu et al. [47] implemented a 12-layered deep 1D convolutional neural network for the precise identification and extraction of five arrhythmia micro-classes. The model achieved a classification accuracy of 97.40%; however, due to the deeply layered architecture, the overall training time of the model increased to 120 min while requiring a further 11 h for the ten-fold cross-validation to complete. The benchmark with the other approaches was based on a few performance metrics such as accuracy, sensitivity, specificity, etc. An excessive amount of preprocessing was carried out to organize the data for a controlled supervised learning environment. The approach portrays no such drawbacks in adaptation except the increased computational time which ultimately increases the cost of the approach as a whole.

The rundown of the literature is depicted in Table 1. The corresponding section sheds light on the advancements and progress, in the context of arrhythmia heartbeat classification, with modern machine- and deep-learning strategies. The literature concludes the importance of deep learning in applications where automated feature recognition reduces the overall computational cost while providing increased accuracy and precision. Toward the development of a scalable, robust, and efficient heartbeat classification model that effectively handles large pools of data, this work proposes an ensemble of deep-learning models. We designed two deep-learning models and merged them for enhanced feature recognition and extraction to classify different types of arrhythmia heartbeats.

## 3. Proposed Methodology

The proposed framework is an ensemble of two deep-learning models: CNN and LSTM, as illustrated in Figure 1. The selection of both models for the ensemble was based on their exceptional performance in automated feature extraction and recognition as depicted in the literature. Since 1D CNNs are impervious to the time-step order, several hidden layers are stacked to extract longer sub-sequences of data that enhance the recognition rate of the model. Meanwhile, since the used datasets accommodate complex ECG signals, LSTM automatically extracts the timing characteristics of these signals which the CNN may miss. Furthermore, the merging of the models requires at least two models in the architecture which can be increased up to *n* number of networks, depending on the computational resources available at hand. Once the models are designed and compiled, they are then merged to form a single multi-layered structure. This methodology combines all the layers in a hierarchical manner that effectively extracts the most optimal features from each layer, fuses them, and forwards them to the next layer for further processing. Both models have been integrated with batch normalization [48] to standardize and normalize the input of each layer. This reduces the load on the neural network and avoids overfitting. After the merged model is trained, the performance is evaluated on the publicly available dataset. Further subsections incorporate the nuances of the proposed framework:

### 3.1. 1D—Convolutional Neural Networks (CNN)

The first model incorporated in the framework is a 1D CNN which is composed of three hidden layers, one flattening layer, two dense layers (1 fully connected, 1 SoftMax), and a batch normalization layer. The CNN takes as input a “n×1” matrix. The detailed architecture of the CNN is represented in Figure 2. All hidden layers are further comprised of a convolution and pooling layer. There exist various pooling strategies such as min-pool, maxpool, and average-pool, the choice of which depends on the type and quantity of data. In the context of heartbeat classification, the most widely used is the average and maxpool techniques. To that end, we too have adopted max-pooling in the implemented CNN which extracts the largest element from each block of the feature map. Similarly, among various activation functions, Rectified Linear Unit (ReLU) has been integrated into each hidden layer to regularize the model and its parameters (τ).

The batch normalization technique has been adopted to normalize and standardize the input to each layer in the CNN. The final feature matrices are flattened and fed to the dense layer which is distributed into classes by the SoftMax layer. The convolution kernel number in each layer is set as “κ” and the kernel sizes are configured to be “α, β and γ”, respectively. The pooling filter size in each hidden layer is kept as “δ” and a consistent stride of size “s” is adopted. To be able to cover the whole input matrix, the padding for the convolution filter is kept as “same”. All the hyperparameters for the two datasets used in this approach have been summarized in Table 2.

If the height of the convolution filter is considered to be $F_{H}$, the width is $F_{W}$ and the dimension is *d*, then the size of the output of the filter can be defined as (Equation 1):(1)$$F_{M}=\left(H-F_{H}+1\right)\times \left(W-F_{W}+1\right)\times 1$$

### 3.2. Long Short-Term Memory (LSTM)

The second network incorporated in the framework is an LSTM due to its exceptional performance in the extraction of temporal and spatial features. The model is contrived of an input layer, an exclusive LSTM cell, a hidden layer, a normalization layer, and a classification layer as shown in Figure 3.

The LSTM cell is divided into smaller gated units where each gate propagates the flow of information after performing some calculations. The input gate is represented by $C_{t}$ which is responsible for updating the state of the cell. A forget gate ($f_{t}$) controls the flow of information by deciding whether the features are to be propagated in the forward direction or to be removed from the input channel. “tanh and sigmoid (σ)” activations are configured on the input and forget gates that normalize the information (adjusts the values between −1 & 1 and 0 & 1, respectively) before letting them through the gate. The current state of the cell is represented as $C_{t-1}$ which is updated concerning a certain time frame. Point-wise addition and multiplication are completed between various cell states and vectors. The newer hidden states are figured out by the output gate ($O_{t}$) and weights *W* and *U* are assigned to each gate based on their respective configuration. The outputs of the various gates can be represented as (Equation 2).
(2)$$\begin{array}{c} C_{t}=\tanh\left(X_{t}\times U_{C}+H_{t-1}\times W_{C}\right)\\ f_{t}=\sigma \left(X_{t}\times U_{f}+H_{t-1}\times W_{f}\right)\\ O_{t}=\sigma \left(X_{t}\times U_{O}+H_{t-1}\times W_{O}\right) \end{array}$$

The values for the common hyperparameters of LSTM and CNN have been kept the same. The LSTM units in the LSTM cell are configured to be “ζ”. Adam optimizer has been configured to handle the sparse gradient and to optimize the memory usage. “Categorical  Crossentropy” loss function has been used to evaluate the distinction between multi-class probability distributions. LSTM hyperparameters are shown in Table 3.

### 3.3. Merger Module

After the models are defined and created, they are merged in the merger module. The merger module takes two parameters as input from the models, the data to be fed to each model circumspect of its dimensions (the input dimensions of both models should be the same), and all the layers defined in both models. The number of epochs is configured to be “ϵ”. The batch size is kept as “δ”. A graphical portrayal of the merger is depicted in Figure 4. All the compilation parameters for the merged model are shown in Table 4.

The underlying layer in the merged model is the input layer which collects the preprocessed information. The data are initially passed to the hidden layers extracted from the CNN model which outputs the subsets of feature vectors. This subset of features is again processed through the third convolutional layer and in a pipeline hierarchy, the next layer should have been the third pooling layer. However, the output feature vector is combined with the LSTM’s initial input vector in the LSTM input layer. The combined features are then refined through the third max-pooling layer to avoid overfitting and vanishing gradients. The max-pooled features are then processed through the LSTM cell for the extraction of long-term temporal dependencies. The LSTM layer drops the low-level features whereas the high-level features are extracted.

The final set of features is flattened and converted into a 1D array. The batch normalization layer standardizes the input to every layer in the architecture. The matrix-vector multiplication is carried out in the two dense layers (fully connected) and the features are distributed into classes by the SoftMax layer. The working of the merger is represented in Algorithm 1.
**Algorithm 1****: **Merger Module**Require:** X = CNN.compile**Require:** Y = LSTM.compile**Ensure:** CNN.inputDimension == LSTM.inputDimension**Require:** data = CNN.LSTM.input *train, test = data.split* **for**
X←Y
**do**  **merger** = Add()[X.output, Y.output]  **mergedModel** = Model([X.input, Y.input], merger)  **mergedModel**.compile(optimizer, loss, metrics) **end for**  **trainedModel** = mergedModel.fit(*train*)   **testModel** = trainedModel.evaluate(*test*)   {testModel returns the evaluation of the merged model on the testing data} **return** *Classified Heartbeats*

## 4. Experimentation

This paper uses two publicly available datasets for the classification of cardiac arrhythmias. Both datasets incorporate ECG recording against several subjects collected in a controlled environment. Both datasets accumulate distinct feature sets and have been separately accommodated in the experimentation. The details of each dataset are provided in the following subsections:

### 4.1. UCI Arrhythmia Dataset—D1

The UCI Machine-learning repository-based arrhythmia dataset [49] incorporates 452 total instances against 13 types of heartbeats. The term ‘instances’ characterize the records such that each instance represents a distinct record from a separate patient in this particular dataset. Furthermore, 203 instances correspond to male patients whereas 249 are from female patients. Furthermore, a single instance accumulates 279 attributes which include the respective subject’s heart rate, sex, age, PR interval, RR interval, etc., with the last attribute being the type of heartbeat. The class-instance distribution is shown in Table 5.

#### Feature Selection and Train-Test Split

Initially, the missing values in the original data were filled by calculating the mean of all the values for that particular attribute. The attributes with more than 30% missing values were discarded and the resultant features were standardized through a standard scalar unit. The unique patterns (principal attributes) in the features were identified by the Principal Component Analysis (PCA) module and high-level features were extracted as input to the model.

After the feature reduction, the number of attributes was reduced to 50 whereas the number of instances remained at 452. The data were then split into a 60–40 proportion on a record-by-record basis, where a random 60% was separated for training and 40% for testing. The class-instance distribution after the train-test split is shown in Table 6.

### 4.2. MIT-BIH Arrhythmia Dataset—D2

This paper also and mainly uses the publicly available MIT-BIH Arrhythmia dataset [50]. The dataset accumulates 30-min raw ECG recordings of 48 patients of which 25 belong to female patients whereas 22 belong to males. Two records among the 22 come from the same male subject. Each record is composed of dual lead signals where the upper signal corresponds to modified limb lead II (MLII) obtained by the placement of lead on the chest, whereas the lower signal is typically a modified lead V1, in a few cases V2 or V5, and in a single case V4. In another two records, the upper signal was replaced by the modified lead V5 due to surgical constraints that restricted the placement of electrodes on the chest. Each record in the dataset is sampled at a 360 Hz frequency. The upper signal MLII accommodates prominent QRS complexes and the database is mainly divided into five arrhythmia super-classes and 15 sub-classes. This paper classifies five arrhythmia sub-classes, i.e., Normal, Left bundle branch block beat, Right bundle branch block beat, Atrial premature beat, and Premature ventricular contraction. The classes are represented by the annotations N, L, R, A, and V, respectively. The details about the sub-class distribution are provided in Table 7.

Furthermore, the original data in D2 reflects a high class imbalance and the presence of noise against the targeted classes. To address these issues, the signals were initially preprocessed through the denoising and resampling methods. The preprocessing details, respectively, are provided in the subsequent sections.

#### 4.2.1. Noise Removal

The raw ECG signals in the dataset are accumulated with the EMG, power-frequency, etc., interferences which are referred to as noise. For the precise detection of arrhythmia heartbeats, the data must be free from such interferences. To that end, before feeding the data to the model, the raw ECG signals have been de-noised through the Discrete-Wavelet-Transform-based denoising technique [51]. The denoising filter reduces the signal distortion in the QRS complex thus it can be more clearly expressed in the detection of RR intervals.

A sample from the raw ECG signals, before and after noise removal, has been depicted in Figure 5 and Figure 6, respectively.

#### 4.2.2. Data Resampling

Class imbalance can cause the model to be prejudiced regarding the predominant class thus directing to a bad or average classification of the minority class. This exerts a crucial impact on the classification accuracy and other performance metrics as well. The details about the distribution of instances against each class in D2 are provided in Table 8. The dataset employed also suffers from imbalanced instances against the target classes. For D2, ‘instance’ represents each observation against a record such that 74,011 instances of class N refer to 74,011 samples present in the dataset for the following class. Moreover, it can be observed that the total instances against the *N* class far exceed the combined instances of all the other classes, whereas the *A* class accumulates the lowest number of instances. This will cause the model to overfit and be more inclined towards the *N* class. To that end, the proposed model endorses the Synthetic-Minority Oversampling Technique (SMOTE) [52]. SMOTE generates the synthetic-minority class examples and upsample the classes with lesser instances while downsampling the majority classes. In this paper, we upsampled the minority classes (L,R,A,andV) and downsampled the majority class *N* to 10,000 instances each such that each class held a 25% distribution of the overall data as shown in Figure 7. This biased and even distribution and resampling of the classes helped to maintain the integrity of data and reduce the load on neural networks.

Figure 8 depicts the aforementioned sample signal after noise removal, normalization, and resampling.

Furthermore, for the precise assessment of the proposed approach, the preprocessed dataset was divided into a 1:4 proportion where a random 20% of the data were separated for testing and 80% for the training of the model. The dataset incorporates 48 records where each record represents a different patient. The train-test split has been carried out against the overall number of samples after preprocessing (50,000) instead of on a record-by-record basis. The reason for this is that only a few records consume all the targeted arrhythmia classes whereas most of the records accumulate instances against only 2 or 3 target classes. To be biased towards each class, the records with a higher proportion of the dominant class, such as *N*, were pruned. The distribution of resampled instances for training and testing of the model is shown in Table 9.

### 4.3. Performance Metrics

Sensitivity, Specificity, Positive Predictive Value (PPV), and Accuracy have been used to benchmark the performance of modern approaches with the proposed approach. Among these, sensitivity represents the capability of a model to identify the true positives in the data. However, specificity constitutes the capability of the classifier to identify the true negatives among all the negatives. PPV figures out the proportion of the predicted positives among all the actual positives. In addition, accuracy is the measurement that determines whether the model is best at identifying patterns and extracting relations between multiple classes on training and testing data. The formulas for each performance metrics are shown in Equations (Equation 3)–(Equation 6), respectively.
(3)$$\text{sensitivity}=\frac{tn}{tp+fn}$$
(4)$$\text{specificity}=\frac{tn}{tn+fp}$$
(5)$$Ppv=\frac{tp}{tp+fp}$$
(6)$$\text{accuracy}=\frac{tp+tn}{tp+tn+fp+fn}$$
where tp,tn,fp, and fn correspond to the total true positives, true negatives, false positives, and false negatives in the final classification.

### 4.4. Testing Environment

All the experiments in the following research work are performed on the following hardware/software specifications: Intel Core i5-10th generation processor, NVIDIA GTX 1650ti GPU, 16GB Ram, Python 3.1, TensorFlow, and PyCharm 2021.

### 4.5. Model Training

For D1, a time sequence of size 1×50 is fed to the input layer of the CNN and LSTM in batches. Each record in the dataset accumulates 50 attributes, hence indicating the reason for adopting such input sizes. The input is then processed in the initial two hidden layers of the CNN. The reduced features are extracted in the form of a matrix and convoluted in the third layer. Before being pooled for the third time, the merger takes in the initial input again and merges the reduced features with it. The merged input is then pooled and processed through the LSTM cells. The time-domain features are flattened and normalized by the batch normalization layer. The final output is condensed in the two stacked dense layers and heartbeats are classified by the SoftMax layer. The model executes for 500 epochs

For D2, the model takes a batch size of 200 as input. We adopted the batch size of 200 instances as we achieved the best results on it. After the reprocessing, a 1D sequence of instances with size 1×200 was fed to the first input layer which queues the data into the model for the first hidden layer (conv1d−maxpool). 2 convolutional kernels of size 1×8 are applied to the input matrix and new sets of features are extracted. These features are propagated to the first maxpool layer where they undergo pooling. The pooling filter of size 1×2 downsamples the feature set with a stride of 1 along its spatial dimensions and the smaller subsets of high-level features are generated. The smaller subset of features is reprocessed through another hidden layer. The merger breaks the third hidden layer and only stacks the third convolution layer again to perform the final convolutions. The output is then merged with the input of LSTM (1d vector of size 1×200) in the second input layer. The merged sets of features are then pooled in the third maxpool layer and finally processed through the LSTM cell.

The LSTM layer outputs a refined 2D sequence of spatial and temporal features that is converted into a 1D array of high-level features by the flattening layer. The batch normalization layer standardizes the input to all the layers in the model. 2 fully connected layers (dense layers) with 32 and 5 neurons condense the features, respectively. The SoftMax layer predicts the multinomial probability distribution and classifies the heartbeats.

The model executes for 25 epochs and the training time per epoch is recorded to benchmark the computational cost of the proposed approach. Adam optimizer is used to regularize the model and auto-adjust the learning rate after each iteration. Categorical−cross−entropy loss function is used to quantify the errors reported by the merged model.

### 4.6. Results on D1

The proposed approach was initially benchmarked on the UCI Arrhythmia dataset. The accomplishment of the resultant metrics is shown in Table 10. It can be observed that the proposed approach achieves an average sensitivity of 89.11%, specificity of 99.40%, PPV of 91.17%, and classification accuracy of 99.05%. The overall accuracy of the proposed model is calculated to be 93.33%. The low average sensitivity is caused due to the availability of only a few instances against some classes (in some cases only 2 to 4 instances). This caused the network to lack in the ascertainment of patterns and relevance of features. Table 11 shows the confusion matrix of the predicted and actual beats. It can be observed that the label *L* accumulated only two instances in the testing data, whereas the model predicted only one to be from the respective class whereas it incorrectly predicted the other to belong to class AM. This caused the model to generate a sensitivity of only 50% in this particular case. The availability of abundant instances against the minority classes would have generated much better results. No oversampling methodology has been applied to D1 due to the availability of very few overall instances.

Furthermore, the sensitivity of the approach can be improved by accommodating a resampling strategy. However, since the classes in D1 are highly imbalanced, resampling by duplication may cause overfitting thus reflecting an increase in sensitivity while decreasing the accuracy.

### 4.7. Results on D2

The proposed approach was tested to benchmark the robustness of the classifier on the MIT-BIH Arrhythmia dataset. The quantitative assessment was carried out using the aforementioned performance metrics. The accomplishment of the resultant metrics is shown in Table 12. It can be observed that the proposed model achieved an overall classification sensitivity of 98.37%, specificity of 99.59%, a positive predictive value of 98.41%, and accuracy of 99.35%. The confusion matrix corresponding to the classified heartbeats on testing data is shown in Table 13. It tends to be seen that 2019 instances of the *N* class were accurately recognized by the model whereas three were incorrectly labeled as belonging to *L*, 10 to *R*, 60 to *A*, and 21 to *V*. The rows represent the incorrectly predicted instances against the actual classes in the Actual column such that: For the first row, three instances were labeled as *N*; however, the actual label was *L*. Similarly, four instances were predicted to belong to *N*; however, they belonged to class *R* and vice versa.

### 4.8. Comparison with Modern Approaches on D2

The comparison based on the model’s capability to classify arrhythmia heartbeats in terms of accuracy has also been carried out with the state-of-the-art deep-learning works as shown in Table 14. The approaches have been sorted in the earliest approach-first order. The comparison with these approaches has only been carried out with accuracy as the common performance metric. The reason for this is that not all these approaches had benchmarked their performance on the variety of metrics used in this approach.

It very well may be seen that the Deep LSTM model acquired a 95.80% accuracy in the classification of eight sub-classes of heartbeats; however, the model lacked stability when dealing with unlabeled data. The approach required a deeply nested hierarchy of LSTM cells to produce notable results. This ultimately slowed down the training process and increased the computational cost of the approach. The exclusive time-series LSTM classifier managed to classify five arrhythmias with a simple LSTM structure but extensive preprocessing. The approach focused only on the simplicity of the model with emphasis on accuracy (95.00%); however, the evaluation parameters such as sensitivity, recall, specificity, etc., likewise assume a significant part in depicting the robust performance of the classifier. The BiLSTM approach stacked several LSTM layers on top of a bidirectional LSTM and compared the performance with three modified deep LSTM models. The BiLSTM model outperformed the LSTMs in terms of accuracy (95.00%); however, at the cost of considerably extended execution time per epoch. The practicality of the model has a significant dependence on the execution time that cannot be neglected.

The DCNN model used the traditional denoising and resampling techniques, very much similar to the ones employed in the proposed approach, to handle five imbalanced arrhythmia classes. The model trained on the preprocessed data classified the arrhythmias with a 94.00% accuracy, whereas on raw data it showed an accuracy of 93.50%. A lightweight LSTM could achieve the same results with lesser execution time, the approach did not portray any other significant aspect of the classifier. Another approach DNN stacked multiple layers of Artificial Neural Networks (ANN) and fused them with a feature learner module. The module extracted features from unlabeled data and the optimal features were refined with the assistance of the genetic algorithm. The model classified five arrhythmia super-classes with a 94.00% accuracy but did not shed any light on the increased computational cost due to the implementation of the learner module and the genetic algorithm. The 1D-deep CNN achieved a relatively higher classification accuracy of 97.00%, but with an average F1 score of 0.84 (84.00%).

A detailed comparison with the baseline approach, Deep CNN embedded with Ten-Folds Cross-Validation technique (DCNN + TFCV), has been made in the proposed approach as shown in Table 15. The common benchmark parameters were considered to be sensitivity, specificity, PPV, and accuracy. The proposed approach classified the same five arrhythmia sub-classes as the Deep CNN approach and achieved 99.35% accuracy. For a deep comparison, the classification sensitivity and specificity were likewise determined, and the proposed model outclasses the referenced approach not only in average accuracy but in other performance metrics as well. The overall sensitivity, specificity, and PPV of the model after 60 epochs were calculated to be 97.05%, 99.35%, and 97.22%, respectively. The proposed approach returned a 1.32% higher sensitivity rate, a 0.25% increased average specificity, and a 1.19% increased PPV. The proposed approach also effectuated a 1.95% increased accuracy compared to the baseline approach.

To address the issue of increased computational cost due to the excessively longer training times, this paper likewise played out a comparative analysis of the overall execution time between the baseline and the proposed approach on D2. The baseline approach expected roughly 120 min to be done with the training as expressed in the paper. It predicted an additional 11 h to finish the ten-fold cross-validation. In terms of functional execution where real-time information is concerned and speed over accuracy is required, a tedious model dismisses the criticality of data, which could be dangerous at times. To beat such blemishes, the proposed approach engrossed, on average, 17 min to prepare the data and classify the heartbeats (testing) as shown in Table 16. In this unique circumstance, the execution time alludes to the time adopted by a strategy to produce the classification results on the testing data, including the preprocessing and training phases.

## 5. Conclusions

Cardiovascular diseases pose a significant danger to human life and dealing with them strongly relies on the accurate analysis of heartbeats via ECG. Manual analysis of ECG signals by a medical expert is costly in terms of time and resources. To that end, the focus has shifted from manual analysis to automated detection of irregularities in the beating of a heart. This work proposed an efficient automated heartbeat classification framework that accurately classified 13 and 5 arrhythmia heartbeats, from respective datasets, in a cost-effective time. The framework used two deep neural networks in conjunction and merged them in a hierarchical layered structure to form a single robust model. The proposed approach was tested on the UCI Arrhythmia and MIT-BIH Arrhythmia datasets and benchmarked with the state-of-the-art approaches. The comparison of the selected evaluation metrics revealed the superior performance of the proposed approach over modern approaches. A comparison in terms of the execution time was also carried out to exhibit that the approach not only far outclasses the modern works in terms of accuracy, sensitivity, and specificity, but overall model execution time as well. Since the input to both models has been treated as a time series, Raw ECG sequences in the case of D2 and PCA extracted features from D1, we have used CNN and LSTM due to their exceptional performance in ECG classification as signified in the literature. The architecture of the proposed model is general, and the novelty resides in the merger; therefore, the neural networks embedded in the proposed architecture can be replaced by GRU or any other model.

The main limitations of the method include the increased computational cost with the addition of more networks. The method is most likely to fail due to the failure of models incorporated with the merger. That being said, at least one model will produce noteworthy results.

For future prospects, the proposed approach can be enhanced and tested to deal with real-time data. Moreover, the model can be simplified to be deployed on embedded systems. In the recent trends of TinyML [53], micro machine-learning models requiring minimal resources can be deployed on the fog and edge nodes. Software libraries such as “TensorFlowLite” are being used to shrink the deep-learning models to perform on-device analytics without any additional cost.

## Figures and Tables

**Figure 1 sensors-22-05606-f001:**
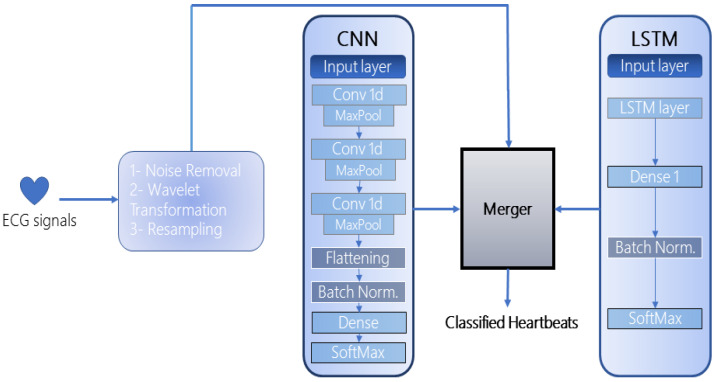
The Proposed Framework.

**Figure 2 sensors-22-05606-f002:**
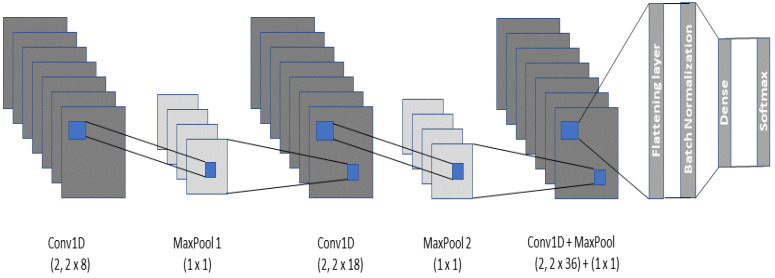
The Architecture of the CNN model.

**Figure 3 sensors-22-05606-f003:**
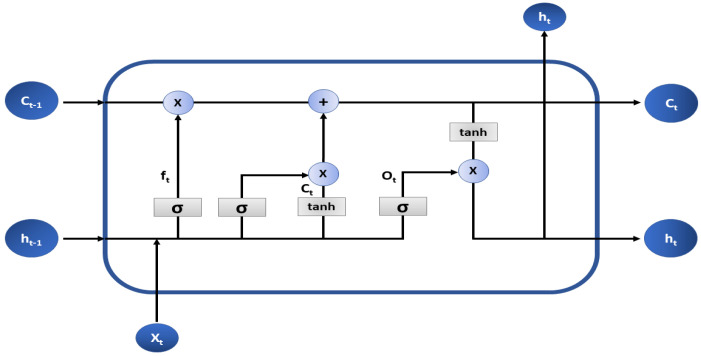
Architecture of the LSTM Cell.

**Figure 4 sensors-22-05606-f004:**
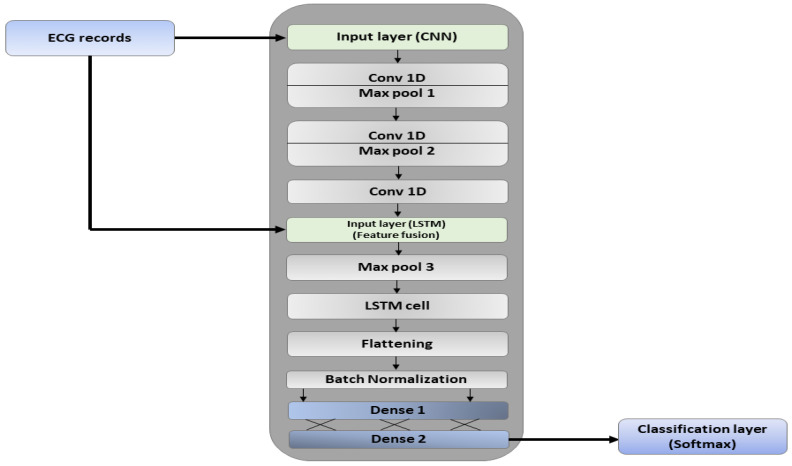
Architecture of the Merger after Compilation.

**Figure 5 sensors-22-05606-f005:**
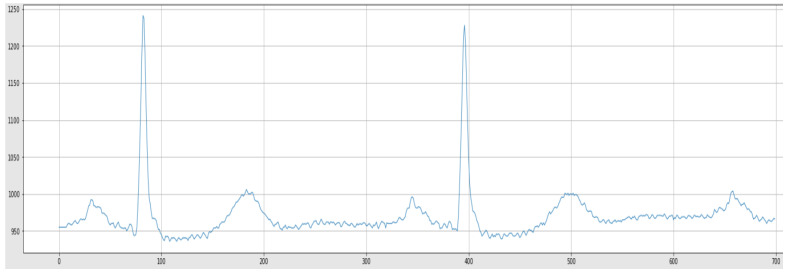
Raw ECG Signal.

**Figure 6 sensors-22-05606-f006:**
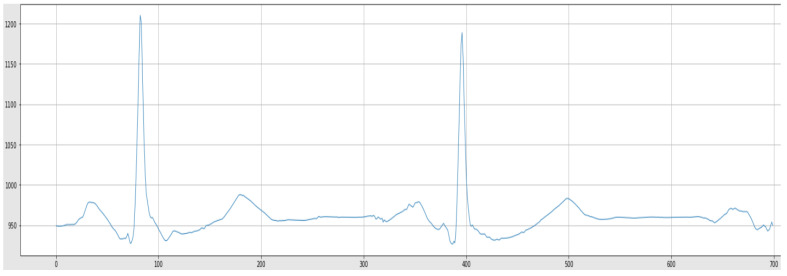
Denoised ECG Signal.

**Figure 7 sensors-22-05606-f007:**
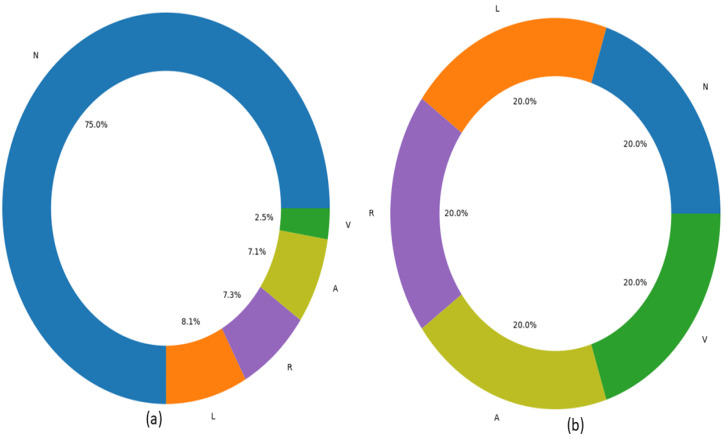
Class distribution (**a**) before resampling and (**b**) after resampling in the dataset D2.

**Figure 8 sensors-22-05606-f008:**
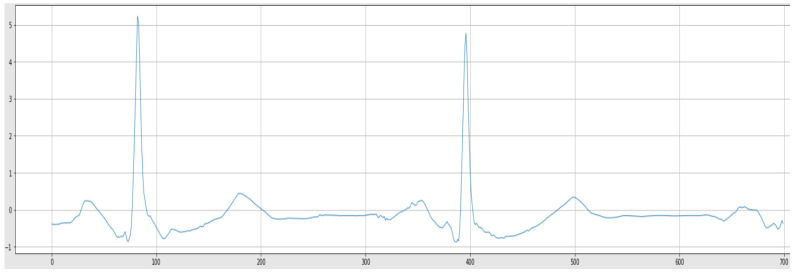
Normalized ECG Signal.

**Table 1 sensors-22-05606-t001:** Literature Overview.

Approach	Algorithm(s)	Accuracy (%)	Arrhythmias Recognized	Limitations
[34]	SVM + RF	77.40	5	A conventional hybrid machine-learning model with average accuracy and high computational cost
[35]	OPF	90.09	5	An efficient classifier that is used in conjunction with other ML algorithms but requires extensive data preprocessing to produce optimal results
[36]	ICCA + ANN	99.60	8	Achieved a very high accuracy on the cost of exponentially longer training time
[37]	SVM + GB	84.82	16	Portrayed average accuracy on a redundant dataset with only 500 records against 16 classes
[38]	Random Forest + BFS	85.58	16	Redundant dataset used with only 500 records against 16 classes
[39]	Echo State Network	98.60	16	Requires too much computational resources leading to a high cost
[40]	LSTM	95.80	8	Model requires longer training time to produce substantial results
[41]	LSTM	95.00	5	Only accuracy considered to be a performance metric, not enough to benchmark an approach
[42]	BiLSTM	95.00	5	A deep LSTM-BiLSTM model which takes too long to train thus increasing the computational cost
[43]	CNN, RNN, Auto-Encoder, DBN	-	16	A survey paper providing an outline with respect to the deep-learning models used in heartbeat classification. No such limitations
[44]	Deep CNN	94.00	5	A deep layered CNN with no such weakness except a deep structure that requires relatively long time to train
[45]	DNN	94.00	5	A DNN combined with a genetic algorithm with high computational cost
[46]	Deep 1D-CNN	97.00	9	A deep CNN with high accuracy but low F1 score
[47]	DCNN + TFCV	97.40	5	Too much training time required considering the amount of preprocessing performed on the dataset

**Table 2 sensors-22-05606-t002:** Hyperparameters for the CNN layers.

Parameter	Value—D1 and D2
Input size	*n*
Stride	*s*
Kernel number	κ
Kernel size—Conv1	α
Kernel size—Conv2	β
Kernel size—Conv3	γ
Pool size	δ
Activation	τ
Padding	same

**Table 3 sensors-22-05606-t003:** Hyperparameters for the LSTM layers.

Parameter	Value—D1	Value—D2
*n*	1 × 50	1 × 200
ζ	64	64
τ	LeakyReLu	ReLu
Optimizer	Adam	Adam

**Table 4 sensors-22-05606-t004:** Parameters for the Merger.

Parameter	Value—D1	Value—D2
ϵ	500	25
δ	25	180
Metric	Accuracy	Accuracy
Loss function	Categorical Crossentropy	Categorical Crossentropy
Optimizer	Adam	Adam

**Table 5 sensors-22-05606-t005:** UCI Arrhythmia Dataset (D1) Class-Instance Distribution.

Super-Class	Annotations	Total Instances
Normal heartbeat	N	245
Ischemic Changes	IC	44
Anterior Myocardial Infarction	AM	15
Inferior Myocardial Infarction	IM	15
Sinus tachycardia	ST	13
Sinus bradycardia	SB	25
Ventricular Premature Contraction	V	3
Supraventricular Premature Contraction	S	2
Left bundle branch block	L	9
Right bundle branch block	R	50
Left ventricle hypertrophy	LV	4
Atrial Fibrillation or Flutter	A	5
Other heartbeats	Q	22

**Table 6 sensors-22-05606-t006:** Training and Testing Data Division—D1.

Class	No. Test Instances	No. Training Instances
N	96	149
IC	14	30
AM	7	8
IM	4	11
ST	6	7
SB	12	13
V	2	1
S	1	2
L	2	6
R	24	26
LV	3	1
A	1	4
Q	9	13

**Table 7 sensors-22-05606-t007:** MIT-BIH Arrhythmia Dataset (D2) Sub-class Distribution.

Super-Class	Annotations	Sub-Classes
Normal heartbeat	N	e, j, N, L, R
Supraventricular ectopic heartbeat	S	a, A, J, S
Ventricular ectopic heartbeat	V	E, V
Fusion heartbeat	F	F
Unclassified heartbeat	Q	f, P, Q

**Table 8 sensors-22-05606-t008:** Imbalanced Instances in D2.

Class	No. of Instances
N	75,011
L	8071
R	7255
A	2546
V	7129

**Table 9 sensors-22-05606-t009:** Training and Testing Data Division—D2.

Class	No. of Test Instances	No. of Training Instances
N	2113	7887
L	2022	7978
R	1967	8033
A	1913	8087
V	1985	8015

**Table 10 sensors-22-05606-t010:** Evaluation of Performance Metrics on D1.

Heartbeat	Sensitivity (%)	Specificity (%)	PPV (%)	Accuracy (%)
N	98.95	97.65	97.22	98.33
IC	86.67	100.00	100.00	98.88
AM	83.33	98.85	71.43	98.33
IM	80.00	100.00	100.00	99.44
ST	100.00	99.42	83.33	99.44
SB	81.82	98.17	75.00	97.14
V	100.00	100.00	100.00	100.00
S	100.00	100.00	100.00	100.00
L	50.00	99.43	66.67	98.33
R	100.00	98.75	91.67	98.90
LV	100.00	100.00	100.00	100.00
A	100.00	100.00	100.00	100.00
Q	77.78	100.00	100.00	98.89
Average	89.11	99.40	91.17	99.05

**Table 11 sensors-22-05606-t011:** Confusion Matrix of the Classified Arrhythmia Heartbeats—D1.

		Predicted												
		**N**	**IC**	**AM**	**IM**	**ST**	**SB**	**V**	**S**	**L**	**R**	**LV**	**A**	**Q**
Actual	N	94	1	0	0	0	0	0	0	0	0	0	0	0
	IC	1	13	0	0	0	2	0	0	0	0	0	0	0
	AM	0	0	5	0	0	0	0	0	1	0	0	0	0
	IM	1	0	0	4	0	0	0	0	0	0	0	0	0
	ST	0	0	0	0	5	0	0	0	0	0	0	0	0
	SB	0	0	2	0	0	9	0	0	0	0	0	0	0
	V	0	0	0	0	0	0	2	0	0	0	0	0	0
	S	0	0	0	0	0	0	0	1	0	0	0	0	0
	L	0	0	0	0	1	1	0	0	1	0	0	0	0
	R	0	0	0	0	0	0	0	0	0	22	0	0	0
	LV	0	0	0	0	0	0	0	0	0	0	3	0	0
	A	0	0	0	0	0	0	0	0	0	0	0	1	0
	Q	0	0	0	0	0	0	0	0	0	2	0	0	7

**Table 12 sensors-22-05606-t012:** Evaluation of Performance Metrics on D2.

Heartbeat	Sensitivity (%)	Specificity (%)	PPV (%)	Accuracy (%)
N	98.54	98.82	95.55	98.76
L	99.56	99.91	99.65	99.84
R	99.13	99.73	98.88	99.61
A	96.03	99.70	98.75	98.98
V	98.60	99.81	99.24	99.57
Average	98.37	99.59	98.41	99.35

**Table 13 sensors-22-05606-t013:** Confusion Matrix of the Classified Arrhythmia Heartbeats—D2.

		Predicted				
		**N**	**L**	**R**	**A**	**V**
Actual	N	2019	3	4	14	9
	L	3	2015	0	1	5
	R	10	0	1945	7	0
	A	60	0	17	1889	1
	V	21	4	1	2	1970

**Table 14 sensors-22-05606-t014:** Overall Accuracy of the Proposed Approach and Referenced Deep-Learning Approaches on D2.

Approach	Overall Accuracy (%)
Deep LSTM [40]	95.80
LSTM [41]	95.00
BiLSTM [42]	95.00
DCNN [44]	94.00
DNN [45]	94.00
Deep 1D-CNN [46]	97.00
DCNN + TFCV [47]	97.40
Proposed Approach	99.35

**Table 15 sensors-22-05606-t015:** Overall Results of the Performance Metrics on D2.

Performance Metric	Deep CNN + TFCV	The Proposed Approach
Sensitivity (%)	97.05	98.37
Specificity (%)	99.35	99.59
PPV (%)	97.22	98.41
Accuracy (%)	97.40	99.35

**Table 16 sensors-22-05606-t016:** Comparison of the Training Time with DCNN+TFCV Approach.

Approach	Training Time(m)
DCNN + TFCV	120
Proposed Approach	17

## Data Availability

Not applicable.

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
