# Peer review of "Heartbeat Classification and Arrhythmia Detection Using a Multi-Model Deep-Learning Technique"

_sensors, 2022, doi:10.3390/s22155606_

Round 1

Reviewer 1 Report

1. The innovation of the article is general. What is the signal after resampling and what is the impact on subsequent recognition?

2. What problems are CNN and LSTM integrating to solve.

3. Try to use visual effects to describe all problems.

4. Gru is an upgraded version of LSTM. Why is it not compared with Gru.

Reviewer 2 Report

The manuscript Heartbeat Classification and Arrhythmia Detection Using a Multi-Model Deep Learning Technique proposes the merging of two deep learning models, one based on CNN and one on LSTM approaches, to efficiently classify the UCI Arrhythmia and the MIT-BIH Arrhythmia datasets.

Major issues:

-The explanation of the structure of the proposed model and of its use need to be more detailed with respect to each of the two presented datasets. 

-The explanation of the dataset preparation needs to provide more details allowing to reproduce your scenario without resorting to the Git-Hub code. Dataset 1 presents with 279 attributes for each record, which are processed through PCA to reduce it to 50 attributes. The underlying ECG time series are summarized by the extracted features. Dataset 2, instead, presents with raw dual lead ECG time series, a quite different scenario. How is the input to the two proposed models organized in each case? Are the 50 features from D1 considered as a sequence for the LSTM input? please provide more detail.

-In the light of the above-mentioned difference in the datasets, considerations regarding the use of a similar model for both of them are required. Was the goal of the manuscript that of proposing a general architecture for facing ECG classification problems?

-All Figure and Table captions need to be more detailed.

Minor

-Figure 1: Please review carefully this figure, as it should explain as much as possible about your proposal. I believe a link between the Input layer and each of the two networks is needed.

-Figure 2: Please provide details about the CNN model in terms of filter numbers, input sizes etc.

-Figure 3 is superfluous.

-Par. 4.2.2 Please define the concept of instance and clarify how these lead to the numbers presented in Table 8 given the description in par. 4.2, the sampling rate and the details provided.

-Please separate the presentation of the results from their discussion in comparison with other models, e.g. paragraph 4.7

section 5. Conclusions: What are CVDs?

p. 2 par. 3 The sentence starting with "The tailing drawbacks... " should be rewritten, its meaning is obscure to me.

p.4 par.2 l.4 reveled should probably be "revealed"

p.4 par.5 l.5 acquiring should probably be "requiring"

p.5 Table 1 rightmost column, last row. [..] training time recorded should probably be "required"

p.6. l. 2 "later" should be "layer"

l. 4 take should be takes

p.5 par.2 l.9 I suppose "optimal features" is enough, please delete "most"

etcetera

Reviewer 3 Report

This paper presents a deep learning method for cardiac arrhythmia detection. Due to the interest of the problem that it solves, I find the work of utility for the scientific community. In this sense, I think that it could be suitable for publication in the Sensors journal provided that the following comments are implemented within the document: 

- The advantage of the proposed model in terms of less training time should be quantified with respect to other solutions (a comparative table would be welcome).

- Which are the main limitations of the method?

- Under what conditions is the method most likely to fail?

- How could sensitivity of D1 be enhanced?

- Please fix the Table 1 and 2 numbers in pages 14 and 16, respectively.

- At least the distribution of men and women in D1 should be specified, as in D2.

- I am not sure about the usefulness of Table 9, since all values are the same.

- The text in Figure 5 should be enlarged for better visualization.

Round 2

Reviewer 1 Report

Accepted without opinoin.

Author Response

We thank the Reviewer for accepting the paper.

Reviewer 2 Report

The paper was improved, yet not all my previous observations are dealt with.

Concern #8 regarding the concept of instance as input to the network still needs an explanation. On  p. 12 the fact that in the database there are 245 instances was already clear, but not how these become the thousands that are downsampled to 10000 just a few lines later. The resampling and class balancing is performed on numbers that cannot be understood. What is an instance as considered in Tables 8 and 9? One sequence of 200 samples as explained in par. 4.5? If so this information need to be available before dealing with balancing.

Again, please explain what is an instance as considered as input to the network. These paragraphs 4.2 through 4.5 need to explain to the reader the flow of processing of the data provided as input to the networks, please rewrite and reorder.

Par. 3.3 Merger Module. "However the output feature vector is combined with the LSTM’s initial input vector in the LSTM input layer. The fused features are then refined through the third max-pooling layer to avoid overfitting and vanishing gradients." Please explain what "combined" and "fused" mean, were they numerically added as algorithm1 hints?

Regarding the modified figure 2, I suggest indicating the symbols in table 2 instead of numbers.

p. 6 l.6 "the merging of models requireS", needs an "s"

p.14 par.2 "For D2, the model takeS", needs an "s"
